# Empirical estimate of the signal content of Holocene temperature proxy records

Maria Reschke[1,2], Kira Rehfeld[1,3], Thomas Laepple[1]

[1]Alfred Wegener Institute Helmholtz Centre for Polar and Marine Research, Telegrafenberg A45, 14473 Potsdam, Germany
[2]Institute of Earth and Environmental Sciences, University of Potsdam, Karl-Liebknecht-Str. 24/25, 14476 Potsdam, Germany
[3]Institut für Umweltphysik, Ruprecht-Karls-Universität Heidelberg, Im Neuenheimer Feld 229, 69120 Heidelberg, Germany

*Correspondence:* M. Reschke (mreschke@awi.de)

**Abstract.** Proxy records from climate archives provide evidence about past climate changes, but the recorded signal is affected by non-climate related effects as well as time uncertainty. As proxy based climate reconstructions are frequently used to test climate models and to quantitatively infer past climate, we need to improve our understanding of the proxy records' signal content as well as the uncertainties involved.

In this study, we empirically estimate signal-to-noise ratios (SNRs) of temperature proxy records used in global compilations of the mid to late Holocene (last 6000 years). This is achieved through a comparison of the correlation of proxy time series from close-by sites of three compilations and model time series extracted at the proxy sites from two transient climate model simulations, a Holocene simulation of the ECHAM5/MPI-OM model and the Holocene part of the TraCE-21ka simulation.

In all comparisons, we found the mean correlations of the proxy time series on centennial to millennial time scales to be low (R < 0.2), even for nearby sites, which resulted in low SNR estimates. The estimated SNRs depend on the assumed time uncertainty of the proxy records, the time scale analysed, and the model simulation used. Using the spatial correlation structure of the ECHAM5/MPI-OM simulation, the estimated SNRs on centennial time scales ranged from 0.05 - assuming no time uncertainty - to 0.5, for a time uncertainty of 400 years. On millennial time scales, the estimated SNRs were generally higher. Use of the TraCE-21ka correlation structure resulted generally in lower SNR estimates than for ECHAM5/MPI-OM.

As the number of available high-resolution proxy records continues to grow, a more detailed analysis of the signal content of specific proxy types should become feasible in the near future. The estimated low signal content of Holocene temperature compilations should caution against over-interpretation of these multi-proxy and multi-site syntheses until further studies are able to facilitate a better characterisation of the signal content in paleoclimate records.

# 1 Introduction

Improving our understanding of the climate systems and its variability requires knowledge about the climate of the pre-instrumental period. Proxy records from different climate archives are available for determining past climate conditions (e.g., Bartlein et al., 2011; Huguet et al., 2006; Johnsen et al., 2001; Li et al., 2006; Luckman et al., 1997). However, as any observational estimate, paleoclimate proxies are affected by uncertainties (e.g., Breitenbach et al., 2012; Lohmann et al., 2013).

The signal that can be retrieved from paleoclimate archives record depends on various temporal (seasonal recording, dating), geological (mixing, transport, sorting), biological (life-time of organisms, habitat depth, bioturbation), and chemical (preservation and dissolution) processes (e.g., Bard, 2001; Berger and Heath, 1968; Goreau, 1980; Leduc et al., 2010; Lohmann et al., 2013; Mollenhauer et al., 2003; Ohkouchi et al., 2002; Rehfeld et al., 2016; Rosell-Melé and Prahl, 2013; Schneider et al., 2010; Telford et al., 2004; van Sebille et al., 2015).

Therefore, the proxy variations do not only contain the climate signal of interest (e.g., annual mean temperature), but also other climatic influences as well as non-climate variability. This poses a challenge to the interpretation of proxy signals, especially in systematic model-data comparisons and quantitative data synthesis efforts. Different approaches have been proposed in an effort to alleviate this problem and improve analyses:

(i)     obtain a better statistical or mechanistic understanding of how and what a proxy actually records (e.g., Fisher et al., 1985; Grauel et al., 2013; Ho and Laepple, 2016; Münch et al., 2016, 2017; Richey et al., 2011; Rosén et al., 2003; Thirumalai et al., 2013);

(ii)    modelling of the proxy signal (e.g., Dee et al., 2011, 2015; Dolman and Laepple, 2018; Evans et al., 2013; Roche et al., 2018); and

(iii)   through a detailed, expertise-driven analyses of single sites (e.g., Stebich et al., 2015).

In this study, we use a comparison of proxy records and model simulations to improve the characterisation of proxy uncertainties through empirical estimates of the signal-to-noise ratio (SNR) in temperature-related proxies. At present, studies on SNRs in proxies are rare and mainly focussed on the instrumental period (e.g., Mann et al., 2007, 2008; Smerdon, 2012, Münch and Laepple, 2018). In contrast, the present study focusses on the pre-instrumental Holocene period which has received considerable attention in the community (e.g., Bakker et al., 2017; Gajewski, 2015; Mangerud and Svendsen, 2018; Marcott et al., 2013; Mischel et al., 2017; Moossen et al., 2015; Sejrup et al., 2016; Thibodeau et al., 2018; Wanner et al., 2015; Zhang et al., 2017). In particular, we focus on estimating SNRs in temperature-sensitive proxy records to improve analyses of Holocene temperature evolution and variability. A better understanding of Holocene proxy time series SNRs will lead to improved and more reliable interpretation of proxy records in multi-proxy and multi-site data compilations and should raise awareness for the need of careful and critical evaluations of paleoclimate reconstructions.

## 2 Data

This study builds on existing compilations of recalibrated high-resolution Holocene temperature-sensitive proxy records to facilitate intercomparison of multiple time series. The analysis is based on three proxy datasets and two model simulations to test the robustness of our results and the sensitivity to the choice of a particular climate model.

### 2.1 Proxy records

We focus on globally distributed multi-archive and multi-proxy compilations of the Holocene temperature evolution from a wide variety of locations (Fig. 1a-c, Tab. 1; Tab. S1-S3), namely

(1) M13: the compilation of Marcott et al. (2013) that was used to reconstruct the global and regional temperature evolution of the past 11.3ky;

(2) LH14: Uk37 and Mg/Ca proxy data compiled in the extended dataset of Laepple and Huybers (2014a) that was used to reconstruct regional temperature variability and builds on the compilation of Leduc et al. (2010);

(3) R18: the Holocene part of the compilation of Rehfeld et al. (2018) that was used to compare Glacial and Holocene temperature variability.

The datasets mostly originated from marine sediment cores and the proxy types include Uk37, planktonic foraminifera Mg/Ca, $TEX_{86}$, terrestrial bio-indicators (fossil pollen modern analogue technique, fossil chironomid transfer function), ice-core stable isotopes ($\delta^{18}O$, $\delta^{2}H$) and several others. As the early Holocene was influenced by deglaciation following the Last Glacial Maximum (e.g., Kaplan and Wolfe, 2006), we restricted the time series to the last 6ky (6ky BP to present day, where BP denotes years before 1950). We only analysed time series containing climate information on at least centennial to millennial time scales (i.e., a mean inter-observation time step of $\Delta t < 500y$). Due to the limited number of available high-resolution time series, the datasets overlap (Tab. 1) to some degree and are thus not independent.

### 2.2 Climate model simulations

We analysed surface air temperature data from simulations of two coupled atmosphere-ocean general circulation models: A 6ky transient Holocene simulation from ECHAM5/MPI-OM (henceforth abbreviated as MPI6k) (Fischer and Jungclaus, 2011) and the TraCE-21ka (T21k) (Liu et al., 2009) simulation from the CCSM3 model, both of which have been used frequently in recent studies (e.g., Gregoire et al., 2016; Heinemann et al., 2009; Koldunov et al., 2010; Lu et al., 2018; Matei et al., 2012; May, 2008; Müller and Roeckner, 2008; Pausata and Löfverström, 2015; Werner et al., 2016). For the present study, annual means of temperatures of both model simulations were extracted at the nearest grid-box related to the proxy record locations of M13, LH14 and R18. Our choice of annual means is consistent with the standard interpretation of these multi-proxy datasets to represent annual mean temperatures (Marcott et al., 2013). This interpretation is a pragmatic choice motivated by the lack of accurate information about the proxy and location-specific seasonality across all records forming such a multi-proxy dataset.

MPI6k (Fischer and Jungclaus, 2011) is a 6ky transient run using ECHAM5/MPI-OM (Jungclaus et al., 2006) which consists of the atmosphere component ECHAM5 (Roeckner et al., 2003), the ocean component MPI-OM (Marsland et al., 2003), and the land surface model JSBACH (Raddatz et al., 2007) with dynamic vegetation module (Brovkin et al., 2009). The model outputs atmospheric variables on a regular longitude/latitude model grid with 96 by 48 horizontal grid-boxes (T31 resolution corresponding to 3.75° in latitude/longitude). The simulation is forced only orbitally with greenhouse gas concentrations set to pre-industrial values. We extracted annual mean surface temperatures at an elevation of 2m from this model (model variable temp2).

The TraCE-21ka dataset (Liu et al., 2009) is originated from a simulation of the transient climate between 22ky BP and 1990 CE and based on a fully-coupled CCSM3 simulation with an atmospheric resolution of T31_gx3 (96 by 48 horizontal grid corresponding to 3.75° in latitude/longitude). Transient forcing factors in the time-period analysed here (last 6ky BP) are changes in the orbitally driven insolation, greenhouse gas concentrations and the meltwater fluxes for the southern hemisphere in the period earlier than 5ky BP.

Our analysis is independent of the absolute changes and only relies on the simulated spatial correlation structure. For the time scales analysed and the proxy positions of our compilations, this correlation structure is not sensitive to the particular choice of temperature variable (sea surface temperature versus surface temperature or near surface air temperature) in either model.

## 3 Method

### 3.1 Approach and assumptions

SNRs can be estimated by comparing proxy records that experienced the same or very similar climate signals, e.g., different proxies from the same site or the same proxy from different sites in close spatial proximity. If a pair of records contains the same signal, an independent local noise component, and no time uncertainty, the SNR is given as $R/(1 - R)$ where R is the correlation between both time series (Fisher et al., 1985). Ideally, SNRs would be estimated from local replicates. This is often difficult, or impossible, due to the limited availability of replicated datasets. To increase the number of records and thus improve the robustness of estimates, we extended this approach to also include records from locations that are further apart. This increased spatial separation between sites requires knowledge of the signal covariance (as the climate signal will have been slightly different at each location) and we rely on climate models to provide this information.

The underlying assumptions are thus: (1) when relying on model data, we must assume correctness of the model-based correlation structure; (2) when using different proxies, we must assume that all proxies recorded the same temporal (and spatial) variability of the climate signal (more specifically: annual mean surface temperature); (3) we must assume that differences in the spatial correlation structure between models and proxy observations are due solely to a site-independent additive noise and time uncertainty. With assumption (2) we discount the seasonality of proxies in this study, but discuss the effects of this strong assumption in section 5.3.

Based on these assumptions, we can estimate the SNR by matching the spatial correlation of proxy records and model time series while accounting for time uncertainty and additive noise which can both lead to a deterioration in the spatial correlation. E.g., low correlations among time series can be caused by both: a low time uncertainty in combination with a high noise level and a high time uncertainty in combination with a high SNR (low noise level). Due to this relationship, we

quantify SNR estimates as a function of time uncertainty.

Sites that are very far apart only share a weak climate signal which does not represent any constraint on the SNR as both the climate and proxy correlations will be close to zero. For our SNR estimate, we therefore only included proxy pairs with spatial separations of up to 5000 km, which we found to be a typical decorrelation distance on centennial time scales in the model simulations as we later show.

As climate variability is a function of time scale, we expect that both the spatial correlation structure and SNR will also be time scale dependent. However, the limited number of records and samples in each record prevents a more thorough time scale dependent estimate which could be carried out using a spectral approach for instance (Münch and Laepple, 2018). In order to balance accounting for time scale and estimate robustness, we distinguish between a centennial time scale $T_{cent}$ (with a cut-off frequency of 1/400y and by removing the linear trend of the time series) and a centennial to millennial time

scale $T_{mill}$ (using a cut-off frequency of 1/1000y and including the trend). To estimate $T_{cent}$, we only used records with a mean sampling interval of less than 200y while all records were included for estimating $T_{mill}$.

**3.2 Spatial correlation structure of model vs. reanalysis data**

As our study depends on the model-based correlation structure, we first analyse this correlation structure at the grid cell level by fitting an exponential, $R = e^{-x/l_d}$, to the decay of correlations R as a function of site separation x for time scales larger

than 400y with included trend (Fig. 1d, e). We further compare the simulated spatial correlation structure with the spatial correlation structure estimated from reanalysis data using the same method. For this aim, we analyse the annual mean surface temperature field of the 20C3M reanalysis (Compo et al., 2006) (Fig. 1f).

Analysing the entire reanalysis period from 1871 to 2011 results in a high estimate of the mean correlation decay length $l_d$ (~9150 km) that is considerably larger than the correlation decay length found in the MPI6k Holocene model simulation

(~2240 km) when analysing the same time scale (unfiltered annual data) as for the reanalysis data. Reducing the human influence (i.e., anthropogenic forcing) by analysing 1871 to 1950 reduces the correlation decay length and results in a similar estimate (~3020 km) than the annual estimate from the MPI6k Holocene simulation (Fig. S1). This result indicates that the model correlation decay lengths used in this study are not unrealistically large and the larger centennial (Fig. 1) than interannual (Fig. S1) decay lengths are consistent with the expectation that temperature fields on longer time scales are more

spatially coherent (e.g., Jones et al., 1997; Kim and North, 1991). The general similarity between the model correlations and the correlations in the reanalysis data also holds when we only compare the correlation between the proxy sites (Fig. 3), suggesting that similar conclusions could be also drawn when using the reanalysis correlation structure instead of the model simulations.

### 3.3 Processing steps

### 3.3.1 Estimation of the spatial correlation structure

From the MPI6k and T21k model time series we extracted annual mean temperatures at those grid cells that contain the location of the proxy record site. As our aim is deriving a time series from the annual model time series that resembles the

proxy time series in having the same number and ages of the proxy observations, we apply block averaging. To get a data point for the observation time $t_i$ we average all observations between half the difference to the previous observation time $(t_i - \Delta t_i/2)$ and half the difference to the next observation time $(t_i + \Delta t_{i+1}/2)$. We chose to use averages rather than interpolation because sediment and ice samples, in particular, often include adjacent depths or have a sample distance that is smaller than the typical mixing depth in the sediment (Berger and Heath, 1968) or diffusion length in ice-cores. For each

proxy compilation (M13, LH14, R18), we estimated the time scale dependent ($T_{cent}$, $T_{mill}$) correlations between all possible proxy record pairs. We further estimated the time scale dependent correlations between all model time series pairs. For this step, the irregularly sampled time series were linearly interpolated onto a regular grid ($\Delta t = 10y$) and subjected to a Gaussian filter with cut-off frequency 1/400y ($T_{cent}$) and linear detrending or alternatively to a Gaussian filter with cut-off frequency 1/1000y ($T_{mill}$) omitting the detrending step. This approach has been shown to deliver good results for the

estimation of time scale dependent correlations in tests using surrogate data with the sampling properties of Holocene marine sediment cores (Reschke et al., 2019).

The spatial separation between two sites was used to place the pair into 2000 km-sized bins (thus containing separations of 0-2000km, 2000-4000km, etc.) and averaging the correlations from proxy/model site pairs contained within the same bin. An overview of the processing steps is given in Fig. 2.

We performed a significance test of the spatial correlation structure based on spatially uncorrelated surrogate time series with a temporal power-law scaling of $\beta = 1$, which is a typical value for Holocene sediment records (Laepple and Huybers, 2014a). In a Monte Carlo procedure with 1000 repetitions, we generated annual surrogate records that were analysed using the same procedure as the true proxy observations, using the 90% quantile of the binned correlations of the surrogate time series as confidence intervals.

### 3.3.2 Estimation of the SNRs

The SNR estimate was obtained from a Monte Carlo simulation with 1000 repetitions. Through block averaging, we resampled the annual model data at the same resolution as the corresponding proxy records. We then added time uncertainties (between 0 and 400y) and noise levels ($0.01 < SNR < 100$), before estimating the mean correlation using the interpolation method of Reschke et al. (2019). We estimated the SNRs as a function of the time uncertainty by minimising

the absolute difference of the mean correlations of proxy records and modified model simulations.

We generated the modified model data by separately distorting the time axis and adding noise to the observations of the resampled model time series. As a simple heuristic to simulate time uncertainty, we defined four time control points at the

ages of 1y, 2ky, 4ky, and 6ky and randomly shifted these points by adding a random value from a normal distribution (mean $\mu = 0$, standard deviation $\sigma$ = time uncertainty [y]) except for the value of 1y. The new time axis was then created by linearly interpolating between the time control points. Noisy observations were generated by adding normally distributed noise, $\varepsilon$ (with $\varepsilon \sim N$, mean $\mu = 0$ and variance $\sigma^2$ as $\frac{\sigma^2_{model,resampled}}{SNR}$), to the resampled model time series. Fig. 2 gives an overview of the processing steps.

# 4 Results

## 4.1 Spatial correlation structure and correlation decay length

The correlation analysis using all proxy types and locations yielded, unsurprisingly, a general decrease in correlation for larger spatial separations between proxy sites (Fig. 3). Both model simulations exhibit statistically significant spatial correlations at both analysed time scales ($T_{cent}$ and $T_{mill}$) and for most inter-site separation distances. Throughout all datasets and separation distances, T21k yielded higher correlations than MPI6k, which is consistent with the generally higher correlation decay lengths $l_d$ for T21k, estimated at grid cell level (Fig. 1d, e).

While for $T_{cent}$ the correlation of both model simulations decreases with increasing site separation (Fig. 3a-c), the $T_{mill}$ estimate (Fig. 3d-f) shows a more complex pattern that includes a partial increase in correlation for separation distances larger than 8000 km. This is likely related to variations in orbital forcing affecting the temperature trend that is partly symmetric (effect of obliquity) and antisymmetric (precession) between both hemispheres. Especially for MPI6k, the correlation is weak for separation distances from 4000 to 6000 km.

The spatial correlations obtained from the proxy records differ systematically from those obtained from model simulation data. The mean correlation for close proxy site pairs (separation <5000 km) was 0.004 to 0.014 for $T_{cent}$ and 0.101 to 0.186 for $T_{mill}$ and thus lower than for model data (MPI6k: 0.303 to 0.338 for $T_{cent}$, 0.202 to 0.461 for $T_{mill}$; T21k: 0.634 to 0.719 for $T_{cent}$, 0.674 to 0.710 for $T_{mill}$). For $T_{cent}$, none of the proxy based correlations is statistically significant and no clear pattern emerges with regard to separation distance. All three datasets yielded a statistically significant correlation at $T_{mill}$ for smaller separation distances, although visibly decreasing for longer separation distances (e.g., 6000-8000 km; cf. Fig. 3d, f).

Comparisons of temperature estimates from different proxy types face the additional challenge that the actual recorded variable (e.g., summer atmospheric temperature vs. mixed layer winter temperature) may depend on the proxy type. We therefore also analysed the proxy-specific results (Fig. 4, Tab. 2). By performing separate analyses for each proxy type (instead of analysing all proxies together) we obtained in all three datasets a higher mean correlation on the $T_{mill}$ time scale for sites within a 5000 km range. For $T_{mill}$, the proxy-specific mean correlations across all datasets and proxies are between 0.149 and 0.357 compared to 0.101 to 0.186 when correlating sites across proxy types. For $T_{cent}$, most correlations are indistinguishable from zero and we observed no consistent increase when analysing proxy-specific correlations (Tab. 2). Unfortunately, restricting the analysis to a single proxy type greatly reduces the number of available proxy pairs at any given

distance and thus leads to less robust correlation estimates and rather large confidence intervals. We therefore only provide results for the most data-abundant proxy types Uk37 and Mg/Ca and one dataset (LH14) as example in the main manuscript (Fig. 4). The remaining data are shown in the Supplement (Fig. S2-S5). For LH14, both Mg/Ca and Uk37 show a decrease in correlation with increasing separation distance for both time scales. The correlations in this proxy-specific analysis are

stronger than the analysis across proxy types (Fig. 3). They are, however, only statistically significant for Uk37 on $T_{mill}$ with separation distances smaller than 5000 km and for a single distance bin (2000-4000km) for Mg/Ca.

## 4.2 SNR estimates

The estimated SNRs of proxy records are a function of time uncertainty because correlations deteriorate due to both, time uncertainty and noise. In general, we found that low (high) SNRs were related to low (high) time uncertainties (Fig. 5). In

most cases, the estimated signal content for Holocene temperature-sensitive proxy records was quite low (<0.5).

By using the spatial correlation structure of MPI6k and assuming a time uncertainty (1 sd) of 220y (mean uncertainty in M13) we obtain an estimated SNR of between 0.05 and 0.2 for the $T_{cent}$ time scale and 0.2 for the M13 and R18 datasets on the $T_{mill}$ time scale. The LH14 dataset yielded a SNR of 10 at the $T_{mill}$ time scale.

For all three proxy compilations (M13, LH14, R18) the SNRs obtained for mixed proxy types depend on the choice of the

model simulation. Using the T21k simulation generally leads to lower SNR estimates ($T_{cent}$: $SNR_{T21k,T_{cent}}$ <0.05; $T_{mill}$: 0.05 < $SNR_{T21k,T_{mill}}$ < 0.2) than using MPI6k as the correlation of spatially close (separation <5000 km) time series pairs is generally higher in T21k. Interestingly, the SNRs estimated using T21k are more similar between the three proxy compilations and thus more consistent than using MPI6k (Fig. S6).

An analysis of proxy-specific SNRs yielded higher uncertainties due to the relatively small number of record pairs and

potentially caused statistically non-robust estimates for some proxy types (see Fig. S7-S16 for the complete set of results and section 5.2 for a sensitivity test of SNR estimates on the number of record pairs). The dependence of SNR estimates on time uncertainty is very sensitive to how the proxies are compiled and the type of model simulation. However, the overview of all proxy-specific SNR estimates (Fig. 6) suggests some proxy-specific tendencies. On $T_{cent}$ ice-cores show the highest SNR. Mg/Ca shows a high SNR for the LH14 dataset but a low SNR in the two other compilations. Uk37 and terrestrial bio-

indicators have the lowest SNR estimate on this time scale. In contrast analysing the $T_{mill}$ time scale that also includes trends in the dataset leads to different results; Uk37 shows the highest SNRs whereas the other proxy types only show a small increase compared to the $T_{cent}$ analysis.

## 5 Discussion

High-resolution temperature-sensitive proxy records for the Holocene are sparse, irregularly distributed, and from different

proxy types. Thus, estimating the SNR in such datasets requires some simplifying assumptions. We assumed that: (1) the spatial correlation of the climate model simulations was realistic, (2) all proxy types were recording the same climate

variable, and (3) any non-climatic components of the proxy signal can be fully accounted for through a combination of time uncertainty and additive noise. As we analysed large multi-proxy and multi-site datasets, in our study we neglected the proxy-specific effects such as seasonality in the recording.

The SNRs we estimated, based on these assumptions, generally suggest a low signal content of Holocene temperature records on centennial time scales ($T_{cent}$). We found a higher signal content on millennial time scales ($T_{mill}$), but the results were rather sensitive to the choice of the proxy compilation and model simulation. We now discuss the role of the different assumptions on the results.

## 5.1 Spatial correlation structure of model simulations

Our SNR estimates critically depend on the model-based temperature correlation structure as lower spatial temperature correlations in the models would lead to higher SNR estimates for the proxies and vice versa. In most regions, the model simulation MPI6k shows correlation decay lengths of 1295 to 6030 km (mean decay length: 3995 km) and the correlation decay length of T21k is generally in the range of 2130 and 8705 km (mean decay length: 5920 km) for time scales larger than 400y with included trend (Fig. 1d, e). This is higher than previous estimates of correlation decay lengths from instrumental datasets in the range of 1000 to 3000 km (e.g., Hansen and Lebedeff, 1987; Jones et al., 1997; Madden et al., 1993). However, such a difference is plausible as an increase with time scale is to be expected. For example, Jones et al. (1997) found lower correlation decay lengths related to annual (2100 km) than to decadal (3800 km) time scales. Indeed, when calculating the correlation decay length for MPI6k on unfiltered annual data, it is consistent to the decay length from instrumental data (Jones et al., 1997) as well as from reanalysis data (Fig. S1).

Nevertheless, spatial correlation could be overestimated in the model simulations for two reasons. Firstly, the spatial correlation of instrumental datasets always includes the anthropogenic forcing which strongly increases the correlation decay length (see Fig. S1 and Jones et al., 1997). This effect is not or only weakly present in the 6ky time-period of our analysis. Instrumental records from the industrial period and pre-industrial model simulations might thus be in agreement for the wrong reasons. Secondly, the grid cell size of the models was on the order of several hundred kilometres whereas the records might be representative of a smaller spatial area. Hence, it is possible that proxy based correlations are lower compared to those obtained from the model due to the former being influenced by subgrid-scale temperature variations. Thirdly, there are several shortcomings in present climate model simulations potentially causing an overestimation of the coherency in the two simulations used in this study. One possibility is that models underestimate internal climate variability that is generally more localised than externally forced climate variability (Laepple and Huybers, 2014a). One mechanism could be a too large effective horizontal diffusivity in the models that would reduce internal variability (Laepple and Huybers, 2014b) and cause larger spatial correlation structures. Further, small scale features and the role of persistent coastal currents might be suppressed by the relatively low, non-eddy permitting resolution of the models used in this study.

We also found that T21k yielded higher spatial correlations compared to MPI6k (Fig. 1d, e) which in turn resulted in lower SNR estimates if relying on this particular model simulation (Fig. S6). This difference might be related to the presence of

transient greenhouse gas forcing in T21k (Timm and Timmermann, 2007), although the changes in forcing were small during the analysed time period.

Thus, there remains the possibility that the true temperature variations are more localised than suggested by the model simulations. In this case our estimates of the proxy signal content would be pessimistic. Ultimately, more replicate proxy records are needed to distinguish between these hypotheses.

## 5.2 Finite number of proxy records

Despite the strong overlap among records, we found our estimates of the spatial correlation structure and SNRs to be sensitive to the choice of proxy compilation (Tab. 2), which suggests that the number of records N may have been limiting the robustness of our estimates. To test this, we performed a sensitivity analysis using different numbers N of surrogate time series ranging from 3 to 50. N 6ky annual surrogate time series were generated from the sum of a common pseudo climate time series modelled as a random process that follows a power-law ($\beta = 1$) scaling and a separate non-climate component that is simulated as uncorrelated white noise. The noise amplitude is chosen to yield SNR = 0.15. Irregular sampling times were used to mimic the observed sampling times of the M13 records. Surrogate inter-observation time steps were drawn from a gamma distribution (shape = rate = 2.25), rescaled with a mean inter-observation time step of 108.56y (cf., Reschke et al., 2019). The final, pseudo proxy time series were then obtained by block averaging the annual time series to the irregular sampling times. The SNR of the surrogate time series were then calculated following the same method as the proxy records in the main study and repeated for different sites using a Monte Carlo-based procedure with 2000 repetitions.

We found that the uncertainty of SNR estimates that are based on a small number of records can be high (Fig. 7). For a low number of only 15 records (105 correlation pairs), for instance, the uncertainty range of SNRs (90% quantiles of 0.08 to 0.26) is higher than the true SNR value of 0.15. Although we used more than 15 sites per compilation in our analysis (Fig. 7), there were often fewer than 15 time series per proxy type (Tab. 1) which might explain the strong scatter in the proxy-type-specific SNR estimates.

To improve the robustness of SNR estimates, it is unavoidable to significantly increase the number of records that are collected not too far apart from one another (distances <5000 km). Additionally, a better global coverage of site locations would likely lead to more robust results. Since we sampled the models at the locations of the proxy sites our results should be independent of the spatial sampling distribution if the models were perfect. In reality, however, spatial differences and shifts in the simulated correlation structure are likely and can be overcome by sampling from a wide variety of sites from all over the globe.

## 5.3 Proxy-specific recording of climate variables

All proxy types used in this study have been reported in the literature as temperature-sensitive and are usually calibrated to the mean annual surface air or surface water temperature. However, this is a gross oversimplification as the true climate variable influencing the recorded signal is proxy-specific and generally more complex. For example, signals reconstructed

from marine organism-based proxies such as Mg/Ca, Uk37, and $TEX_{86}$ are affected by the seasonal and depth-specific preferred habitat of the organism (Ho and Laepple, 2016; Jonkers and Kucera, 2017; Leduc et al., 2010; Lohmann et al., 2013; Tierney and Tingley, 2015). As we currently interpret all records from different proxy types as annual mean surface temperatures, this might influence our results in various ways. Analysing different proxy types with different recording preferences likely leads to an underestimation of the spatial temperature correlations. Indeed, in our study we found the spatial correlations related to records of the same proxy type for $T_{mill}$ to be higher compared to those for all types (Tab. 2). To gain a better understanding of proxies and their effect on the analyses, we suggested to use proxy-specific SNR estimates instead. However, this is currently hampered by the low number of records in close proximity to one another (Tab. 1; Tab. S4). For many proxy types, this leads to statistically non-significant correlations and unreliable SNR estimates (Fig. S7-S16). Additionally, even for one proxy type and proxy carrier (e.g., foraminifera) we expect a site-specific season and depth habitat. Such differences would reduce the correlation compared to the correlation of the climate component sampled at any globally fixed season or depth and would thus bias the SNR estimates low.

Assuming annual mean sea surface temperatures instead of one specific season or depth also influences the correlation structure derived from the models. Calculating the correlation structure of summer and winter in both models (not shown) suggests an increase or decrease of the correlation depending on the choice of the model so that the net-effect on the SNRs is not clear.

Finally, analysing the spatial correlation among records of the same proxy type can also lead to overly optimistic results as the correlation among records of the same proxy type could also stem from spatially correlated proxy-specific non-climatic components. A case in point would be the dissolution of foraminiferal shells (Lea, 2003) which could generate spatially correlated noise as the preferential dissolution of carbonate depends on the water depth (Brown and Elderfield, 1996; Dekens et al., 2002), the carbonate ion concentration, and the salinity of the surrounding seawater (Huguet et al., 2006; Lea, 2003; Spero et al., 1997).

## 5.4 Time uncertainty and non-climatic components of the proxy signal

Our SNR estimates depend on the assumed time uncertainty of the records. While we assumed a mean time uncertainty of 220y (as provided in the M13 dataset), the true time uncertainty for marine records might be considerably higher due to spatially varying reservoir effects (Ascough et al., 2005). This would imply that our SNR estimates are conservative, especially on centennial time scales. On the other hand, using one mean uncertainty value will clearly be too pessimistic for ice-core data that is only subject to much smaller dating uncertainties. Using more sophisticated models to account for time uncertainty (e.g., Blaauw, 2010; Blockley et al., 2007; Telford et al., 2004) and the proxy- and site-specific information on the chronologies would allow to obtain more precise SNR estimates.

We modelled the transfer function between the temperature time series and the calibrated proxy records as a combination of time uncertainty and additive temporally uncorrelated noise. Our approach thus neglects other distortions of the signal and non-additive parts of noise. Multiplicative noise can arise from aliasing due to subsampling that leads to errors that are

proportional to short-term climate variability (Laepple and Huybers, 2013). Variable sedimentation rates, bioturbation and/or bioturbation depths varying over time have a low-pass filtering effect that is similar to irregular sampling. Proxy archive accumulation processes undergo temporal changes due to changes in bioturbation depths and advection (Mollenhauer et al., 2003) or spatial changes in ocean currents (van Sebille et al., 2015) that could introduce additional non-additive noise in the obtained proxy records. Finally, even for a single proxy type, the data quality (i.e., the signal content) is site specific and will depend on the sampling and measurement protocol. For example, the SNR estimates using the LH14 dataset that is mainly based on very high-resolution records (mean sample distance <100y), are higher than estimates based on the two larger proxy compilations. Thirumalai et al. (2018) showed that foraminiferal records based on a large number (70-100) of foraminiferal tests per sample were consistent between cores collected in close proximity to one another leading to much higher correlations compared to our study. As we rely on datasets of opportunity that consist of proxy records measured by various labs over a period of two decades, it seems conceivable that a small number of records could be of a relatively lower quality which would reduce our mean correlation and thus the SNR estimate. New studies, especially when based on a careful design (Thirumalai et al., 2018), could help alleviate this situation.

## 5.5 Implications and future steps forward

Our results underline the challenge of resolving the small Holocene climatic variations in current climate archives, but also challenge the strong spatial coherency of centennial to millennial temperature variations simulated in current climate models. On the proxy side a continuation of the work on understanding the proxy systems is warranted. Examples are the use of modern monitoring systems, sediment traps and culturing studies. Implementing these findings into ecological models of various complexity (e.g., Jonkers and Kucera, 2017; Kretschmer et al., 2018) and proxy system models (e.g., Dolman and Laepple, 2018) is needed to generalise the knowledge and make it usable in global studies. Forward modelling of proxy records will allow estimates of the signal content complementing the empirical estimates provided here. Finally, a better proxy understanding implemented in proxy system models will also allow to optimise the sampling (e.g., sampling and replication strategy) and measurement process (e.g., number of foraminiferal tests). Finally, although labour intensive, a more frequent generation and analysis of replicate records would allow to separate local and climate and non-climate variability and thus provide a key step in understanding the proxy and climate variability as well as the proxy formation process.

Progress in climate modelling is needed to resolve the spatial scales and regions such as shelf areas and coasts sampled by the proxies. Due to the increase in computing power, climate models will be able to perform long (>1000y) and high resolution, often eddy permitting model simulations (e.g., Haarsma et al., 2016). Confronting these simulations with (replicated) sediment records, ideally accounting for the seasonal and depth habitat of the proxy carriers would allow to better constrain the spatial structures of climate variability as well as refining the estimates of the proxy signal content. If our SNR estimates are realistic, Holocene studies relying on a small number of records might be associated with large uncertainties, except if the quality of the analysed records is considerably higher than the average of the records analysed

here. Holocene stacks relying on a large number of records such as used in Marcott et al. (2013) would be robust if the errors are independent across sites. However, extracting spatio-temporal patterns from such datasets will be difficult. If our results are actually too pessimistic, e.g., as the true climate is more regional than simulated by the used model simulation, this would support the current interpretation of individual Holocene proxy records as a regionally representative climate signal. Otherwise, in case of too optimistic SNR estimates, the value of singular Holocene proxy reconstructions without additional expert knowledge would be limited and regional stacks might be needed to extract regional Holocene signals in analogue to the strategy used by the tree ring community.

## 6 Conclusion

In this study, we estimated SNRs of Holocene temperature-sensitive proxy records by comparing proxy- and model-based spatial correlations. We found that spatial correlations between proxy records were significantly lower than those computed for temperature time series extracted from climate models. Simply put, the proxy records varied more independently from site to site, whereas the model simulations suggested spatially coherent temperature variations. This in turn led to low SNR estimates in multi-proxy-type analyses if we assume that the correlation structure that we obtained from the model simulations is reasonable.

The low SNRs of Holocene proxy records are likely the result of processes occurring during the formation, preservation and measurement of the proxy signal. For the Holocene, even small uncertainties in the process chain between the climate signal and the climate reconstruction play an important role compared to the small temperature variations. In addition, as evidenced by the difference when comparing results between proxy types and within one proxy type, the proxy-specific recording of different temporal and spatial parts of the temperature (for example summer vs. winter) also affects the SNR of multi-proxy datasets. Nevertheless, our SNR estimates are still relevant for synthesis and model comparison efforts (e.g., Marcott et al., 2013), that usually interpret all proxy records together. While in the ideal case global stacks based on a large number of records will average out most of the error contributions, the interpretation of spatio-temporal patterns will remain uncertain. The precision of the SNR estimates is strongly dependent on the number of available proxy records. Due to the small number of spatially close records of the same proxy type, the uncertainty in our proxy-type-specific SNR estimates was very high.

Our SNR estimates implicitly depend on the expected time uncertainty, as well as on the model choice. However, for both tested models the multi-proxy-type estimates on centennial time scales ($T_{cent}$) were smaller ($SNR_{MPI6k,T_{cent}} < 0.5$; $SNR_{T21k,T_{cent}} < 0.05$) than on longer time scales $T_{mill}$ ($SNR_{MPI6k,T_{mill}} \approx 0.2$; $0.05 < SNR_{T21k,T_{mill}} < 0.2$).

Our results of the low signal content of multi-proxy and multi-site datasets, especially on centennial time scales, suggests that caution and a critical evaluation are in order when analysing and interpreting such large datasets. Furthermore, optimising the sampling and measurement procedure is likely needed to faithfully reconstruct small climate variations over the Holocene. As the number of high-resolution proxy records continues to grow, a more detailed analysis of the signal

content of specific proxy types and a model-independent estimate of the spatial correlation structure of climate variations will get feasible and will enable and improve prospects for the interpretation and reconstruction of past climate changes.

*Code availability.* Software to reproduce the main analyses presented in this paper is available as an R code under https://link.

*Data availability.* The datasets used in this study are available at www.sciencemag.org/cgi/content/full/339/6124/1198/DC1 (M13), at https://www.nature.com/articles/nature25454#supplementary-information (R18) and at the PANGAEA database (https://www.pangaea.de) under https://doi.org/10.1594/PANGAEA.XXXXX (LH14).

*Author contributions.* M.R. and T.L. designed the research. M.R. performed the analysis and wrote the first draft of the manuscript. M.R., K.R. and T.L. contributed to the interpretation and to the preparation of the final manuscript.

*Competing interests.* The authors declare that they have no conflict of interest.

*Acknowledgements.* We would like to acknowledge Igor Kröner for fruitful discussions and comments on the manuscript. The work profited from discussions at the CVAS working group of the Past Global Changes (PAGES) programme. This study was supported by the Initiative and Networking Fund of the Helmholtz Association Grant VG-NH900 as well as the European Research Council (ERC) under the European Union's Horizon 2020 research and innovation program (grant 15   agreement no. 716092). It further contributes to the German BMBF project PALMOD. K.R. acknowledges funding by the German Research Foundation grants DFG RE3994-1/1 and DFG RE3994-2/1.

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

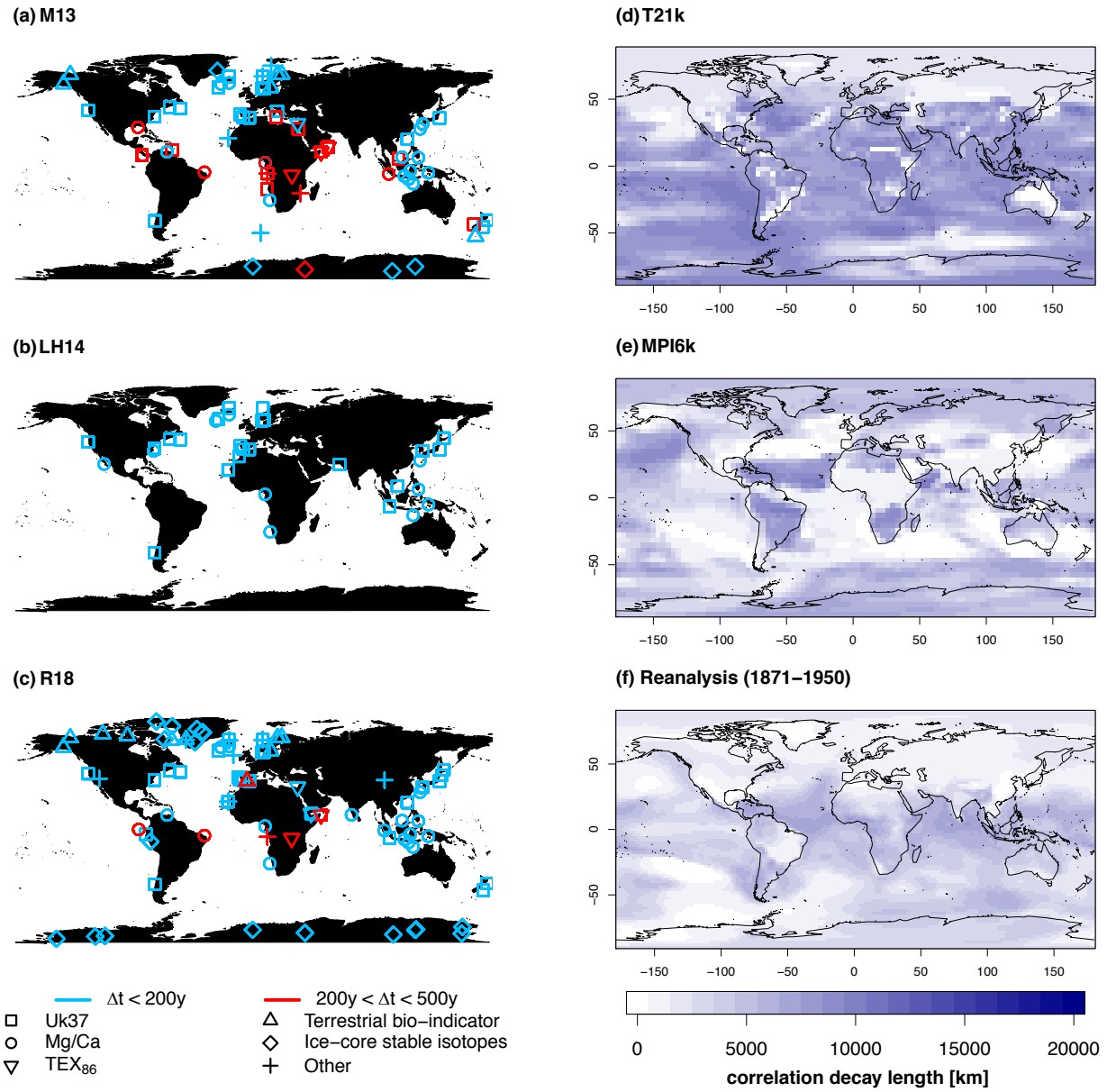

**Figure 1: Overview of proxy and model datasets.** Site locations of the proxy compilations (a) M13: Marcott et al. (2013), (b) LH14: Laepple and Huybers (2014a), and (c) R18: Rehfeld et al. (2018) used in this study. Proxy types are indicated by symbols and the mean inter-observation time step by colours. Correlation decay length of (d) the T21k and (e) the MPI6k simulations estimated on time scales larger than 400y with included trend, and (f) reanalysis data from 1871 to 1950 estimated from annual data. The spatial correlation decay length is generally higher for T21k than for MPI6k. For a comparison of the model and reanalysis correlation structure on the same time scale, see Fig. S1.

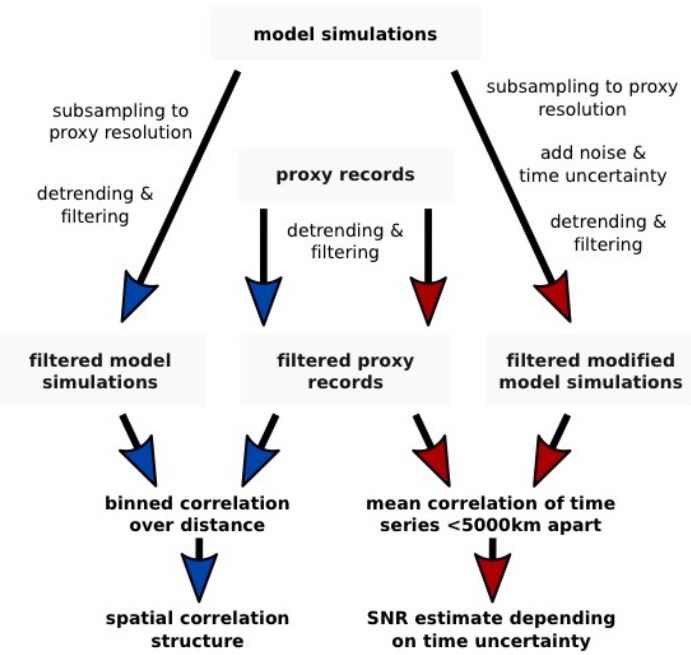

**Figure 2: Processing steps for the proxy and model time series.** Blue paths illustrate the analysis of the spatial correlation structure. Red paths represent the estimation of SNRs of proxy records as a function of time uncertainty.

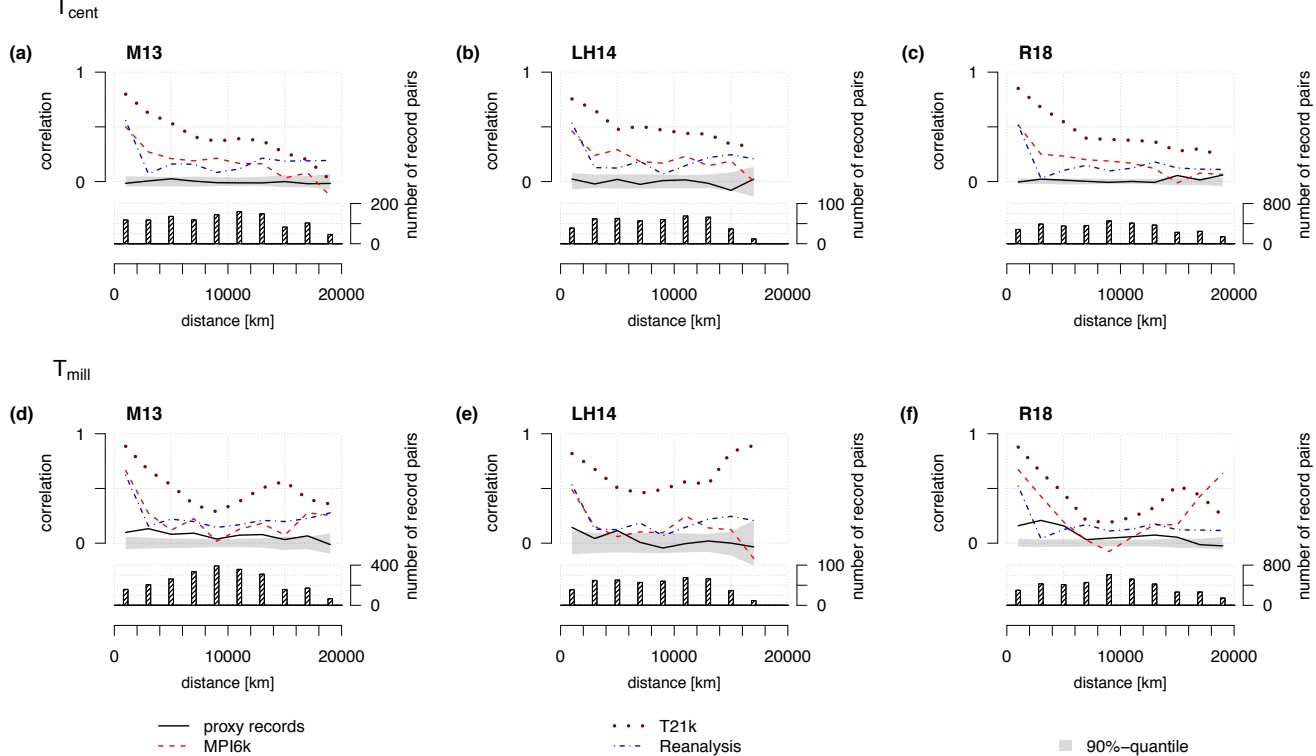

**Figure 3: Spatial correlation structure of Holocene temperature proxy records and simulated surface temperatures based on three multi-proxy datasets and related to (a-c) centennial $T_{cent}$ and (d-f) centennial to millennial time scales $T_{mill}$.** In each panel, the upper part shows the mean correlation of the model simulation (for 2000 km sized bins as a function of the separation distance between record pairs) and reanalysis data (1871-1950) evaluated at the proxy locations (dotted/dashed line) and the proxy dataset (continuous line). The grey polygon represents the 90%-quantile of mean correlations of uncorrelated surrogate time series with a power-law scaling of $\beta = 1$. The lower parts of the panels show the number of record pairs used in each estimate. The spatial correlation structure of the model time series is generally higher than of proxy records which are only statistically significant on $T_{mill}$ at neighboured sites. The highest correlations belong to sites with separation distances less than 4000-6000 km.

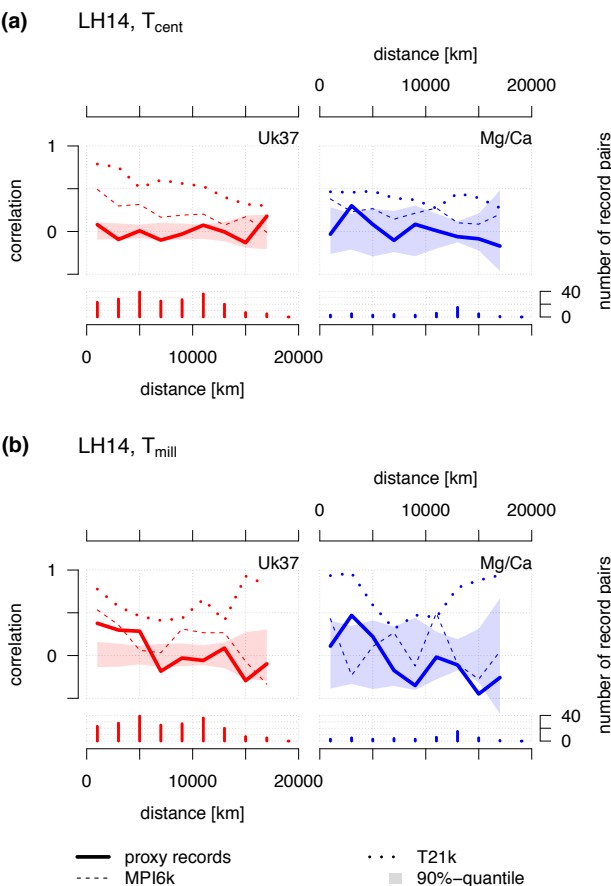

**Figure 4: Proxy-type-specific (Uk37, Mg/Ca) spatial correlation structure related to (a) centennial $T_{cent}$ and (b) centennial to millennial time scales $T_{mill}$ based on the LH14 dataset.** The upper parts of the panels show mean correlations of 2000 km sized bins as a function of the separation distance between record pairs in the proxy dataset (continuous line) and model simulations evaluated at proxy locations (dotted/dashed line). Polygons represent the 90%-quantiles of mean correlations of uncorrelated surrogate time series with a power-law scaling of **β = 1**. The lower parts of the panels show the number of record pairs used for each estimate. The spatial correlation structure of proxy records is non-significant for individual proxy types, except for close (separation <6000 km) sites of Uk37 temperature records at $T_{mill}$.

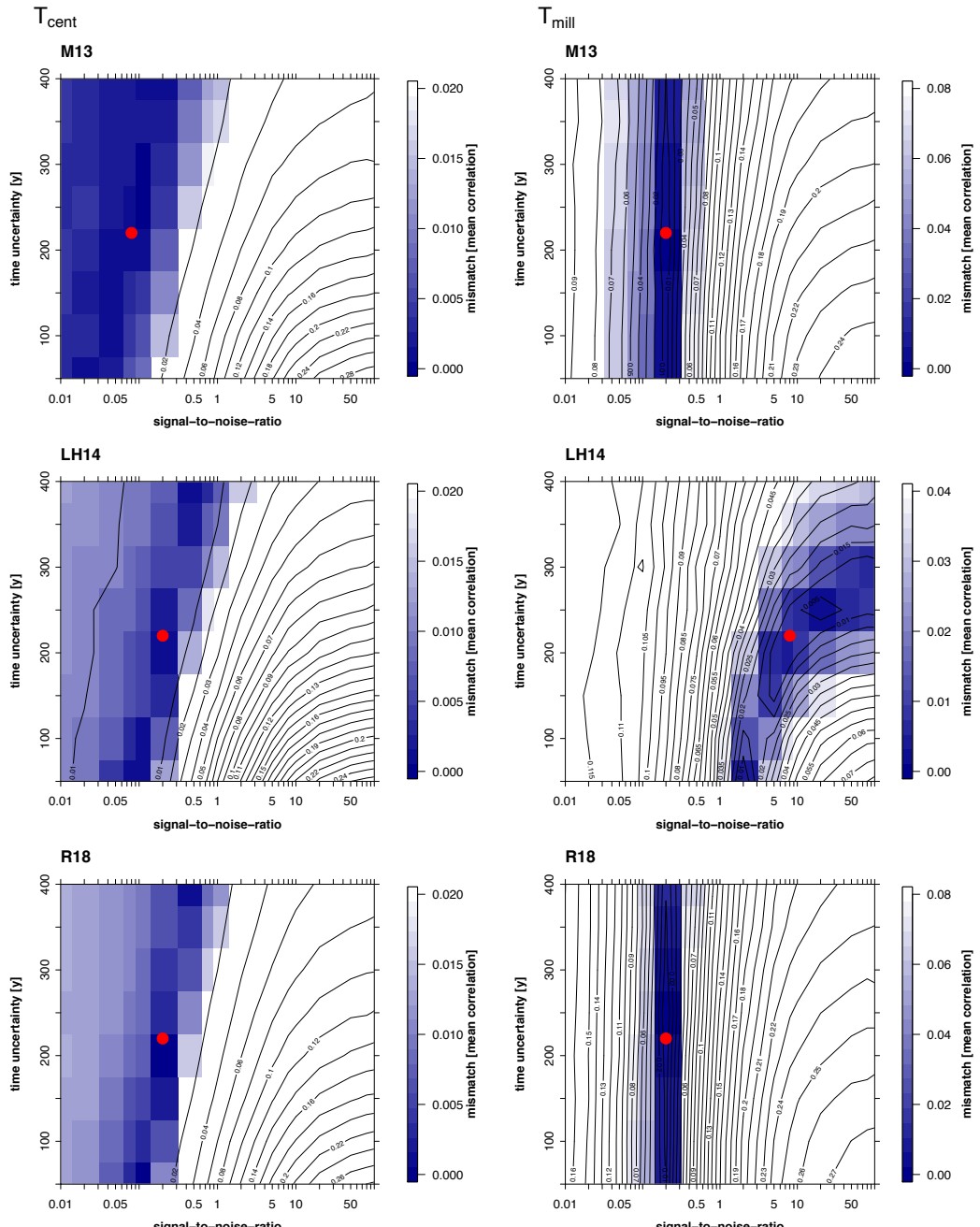

**Figure 5: SNR$_{MPI6k}$ estimates of proxy records as a function of time uncertainty related to centennial T$_{cent}$ and millennial time scales T$_{mill}$.** Colour coating and contour lines in each panel show the mismatch between mean correlations of close-by (separation <5000 km) proxy records and time series extracted from the MPI6k simulation at proxy locations as a function of time uncertainty (vertical axis) and SNR (horizontal axis). Areas with the lowest mismatch are represented by the darkest colours and mark suitable combinations of SNR$_{MPI6k}$ estimates and time uncertainties. The red dots illustrate SNR estimates for a time uncertainty of 220y.

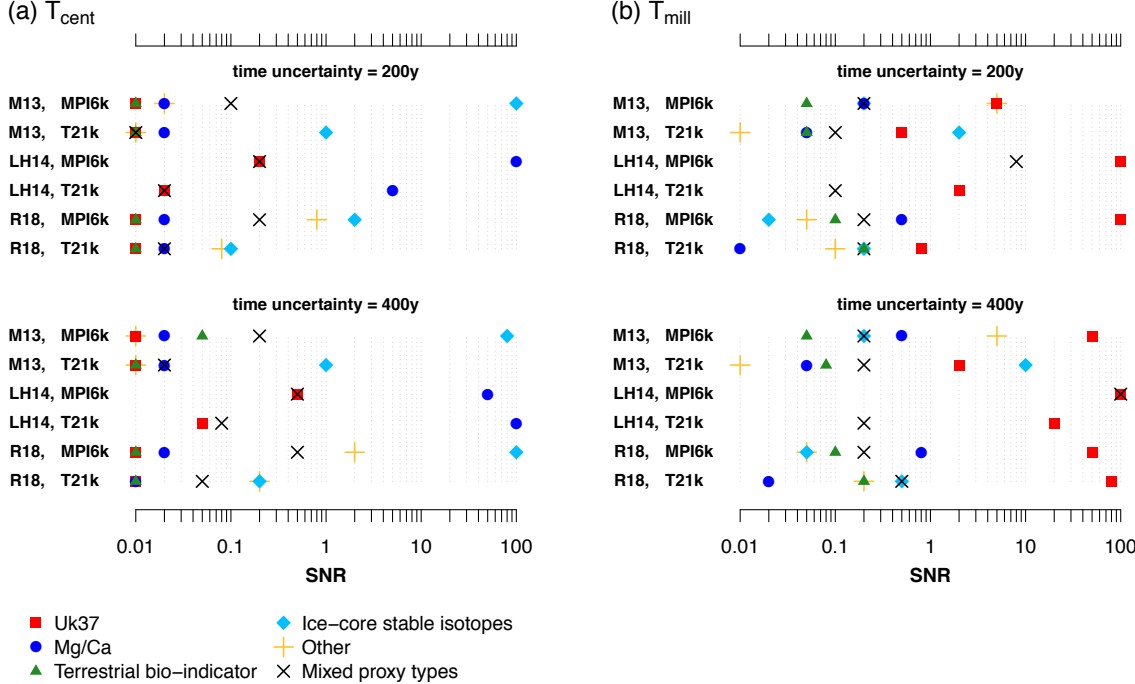

**Figure 6: Overview of proxy-specific SNR estimates on (a) centennial T$_{cent}$ and (b) millennial time scales T$_{mill}$.** The symbols represent the SNRs estimated from the different proxy compilations using the simulations of MPI6k and T21k. Upper panels show the results for an assumed time uncertainty of 200y and lower panels for 400y. The SNRs are proxy-type-specific different, but generally higher on the T$_{mill}$ than on the T$_{cent}$ time scale.

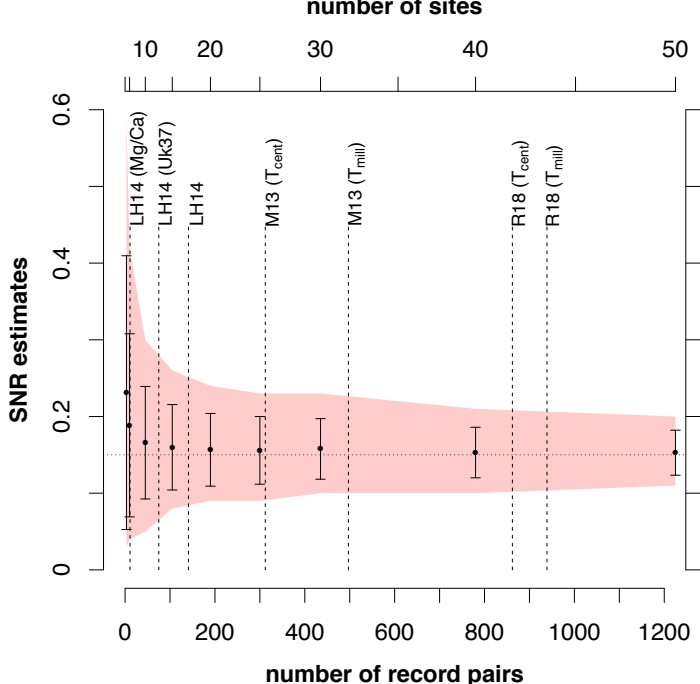

**Figure 7: Sensitivity of the SNR estimates on the number of sites/record pairs based on surrogate time series.** The time series were generated with a predefined SNR = 0.15 (horizontal line). SNR estimates with standard deviations based on 2000 repetitions are shown as
5    dots with error bars. The uncertainties in the SNR are illustrated as polygons showing the 90%-quantiles of the estimates. The uncertainty of SNR estimates is high when only considering a small number of sites. Vertical lines show the numbers of selected sites/record pairs contained in each data compilation. This indicates that for single proxy type analysis the uncertainties in the SNR estimates are high.

**Table 1: Numbers of records and their overlap in the proxy compilations used in this study.** The total number of time series is separated by proxy type for each proxy compilation (upper part). $T_{mill}$ refers to the number of time series with a mean inter-observation time step of $\Delta t$ <500y and $T_{cent}$ counts time series with $\Delta t$ <200y. The overlap is shown for each pair and for all compilations (lower part).

| | Sum of records | Uk37 | Mg/Ca | TEX$_{86}$ | Terrestrial bio-indicator | Ice-core stable isotopes | other |
|---|---|---|---|---|---|---|---|
| M13 − $T_{mill}$ | 70 | 28 | 19 | 4 | 8 | 5 | 6 |
| M13 − $T_{cent}$ | 49 | 18 | 14 | 1 | 8 | 4 | 4 |
| LH14 − $T_{mill}$ | 31 | 21 | 10 | - | - | - | - |
| LH14 − $T_{cent}$ | 31 | 21 | 10 | - | - | - | - |
| R18 − $T_{mill}$ | 88 | 27 | 19 | 4 | 11 | 18 | 9 |
| R18 − $T_{cent}$ | 81 | 26 | 17 | 2 | 10 | 18 | 8 |
| M13 ∩ LH14 | 20 | 13 | 7 | - | - | - | - |
| M13 ∩ R18 | 45 | 16 | 14 | 3 | 6 | 4 | 2 |
| LH14 ∩ R18 | 22 | 16 | 6 | - | - | - | - |
| M13 ∩ LH14 ∩ R18 | 19 | 13 | 6 | - | - | - | - |

**Table 2: Mean correlations of proxy time series with separation distances <5000 km for different proxy types.** For each dataset, the mean correlation was estimated for millennial time scales $T_{mill}$ and proxy time series with a mean inter-observation time step of $\Delta t$ <500y and related to centennial time scales $T_{cent}$ for proxy time series with $\Delta t$ <200y. Mixed proxy types contain all combinations of time series pairs independent of the proxy type. The mean of single proxy types summarises the proxy-type-specific mean correlations weighted by the number of record pairs of each proxy type. Correlations in brackets are not statistically significant (p = 0.1).

| | Mixed proxy types | Mean of single proxy types | Uk37 | Mg/Ca | TEX$_{86}$ | Terrestrial bio-indicator | Ice-core stable isotopes | other |
|---|---|---|---|---|---|---|---|---|
| M13 $- T_{mill}$ | 0.101 | 0.149 | 0.211 | (0.068) | (-0.105) | (0.095) | 0.414 | (-0.14) |
| M13 $- T_{cent}$ | 0.004 | -0.009 | (-0.003) | (-0.006) | - | (-0.09) | (0.244) | (-0.388) |
| LH14 $- T_{mill}$ | 0.12 | 0.357 | 0.365 | 0.304 | - | - | - | - |
| LH14 $- T_{cent}$ | 0.012 | 0.031 | (0.013) | 0.151 | - | - | - | - |
| R18 $- T_{mill}$ | 0.184 | 0.208 | 0.347 | (0.034) | (-0.188) | 0.17 | 0.23 | (0.107) |
| R18 $- T_{cent}$ | 0.014 | 0.009 | (-0.017) | (-0.024) | (0.148) | (-0.057) | 0.1 | (0.05) |