# Peer review of "Empirical estimate of the signal content of Holocene temperature proxy records"

_Climate of the Past, 2018_

## Referee Comment (RC1) · Bothe (Referee) · 10 Dec 2018

Dear authors, dear editor,

First, I have to point to a note on potential conflicts of interest at the end of this review.

**Review:**

I really enjoyed reading the manuscript by Reschke et al. "Empirical estimate of the signal content of Holocene temperature proxy records" (cp-2018-154). It certainly fits the journal Climate of the Past and it is highly relevant for attempts to improve our

understanding of proxy records and climates from the Last Glacial to the Holocene. It helps us to better evaluate the uncertainty of proxy records and reconstructions as well as comparing paleo data and model simulations.

There are two major comments, that I think should be addressed. Most other comments may be seen more as suggestions. Let me first shortly summarise the more important of my major points.

Your results and how much we can infer from them strongly hinges on the assumptions you make. You state those clearly and discuss them already. You do this in your discussion section. From my point of view it is necessary that you extend on these discussions already when you present your assumptions.

I understand that others may disagree with discussions taking place in the methods-section. Indeed it may be that in a follow up review I say, I was wrong, because the manuscript reads better in this version. However considering this version of the manuscript, the lack of a discussion of your assumptions' appropriateness in the method section clouds the reading.

**Major:**

1. Section 3.1: Let me extend on my short summary. First, I think your assumptions are reasonable and well stated and do not invalidate your approach. However, they also can invite strong criticisms. You counter these mostly later in the manuscript, but I think you have to show early that you are thinking about this and why the assumptions are appropriate.

A number of questions you should probably deal with early on are, for example: Aren't models thought to be more homogenous than observations? Isn't it unlikely that the proxies really recorded the same 4D-signal? Less important is possibly, whether you can really capture the uncertainty about the signal in the simple estimates you take.

2. Reference to Reschke et al. (particularly page 6, line 2): Please provide some information in the methods section or at least in an appendix on the method (not least since Reschke et al. is not openly accessible).

**Minor:**

General: Maybe mention your focus on the last 6kyr already in the title or at least in the abstract.

Abstract:

Suggestion: Maybe rethink the abstract to clarify how you proceed (e.g., page 1, line 17 and following). That is, state which simulations you use before mentioning how they influence the results, or don't mention the specific models at all but just highlight the contrasting results.

Suggestion: Mention the use of the correlation structure before discussing its influence.

Introduction:

Page 2, line 9: If I understand your point correctly, this is not only about non-climatic influences but also about climatic influences different from the specific signal we are interested in.

Data:

Page 3, line 12: You possibly should introduce the abbreviation for Methylation of Branched Tetraether - or simply skip mentioning them.

Page 3, line 24: Please discuss, why annual temperature is an appropriate choice.

Page 4, line 3: Are all the mentioned forcing factors really continuously transient in the simulation?

Method:

Page 5, line 8: Can you please provide slightly more information for what this reference is here? I do not directly see how it relates to the sentence.

Page 5, line 13: What do you mean by resampled in this context? Further, could you give more details on your block averaging (e.g., block length).

Suggestion: Page 5, line 16, "For each. . .": I am not sure whether this description gives the reader enough information to redo your analyses. But I am neither sure that it does not. Maybe rethink this.

Page 5, line 20: Please give some more details here on what your reference work describes in this context.

Results:

Suggestion: Page 6, line 25: Would you be willing to discuss how realistic you regard the simulated correlation coefficients, and, possibly, how this may affect the relevance of your SNRs, e.g., if we assume the simulated correlation coefficients are not realistic.

Page 7, line 20: Maybe you should be more explicit in writing about the results for T21k (see also short technical comment below).

Discussions:

Suggestion: I think it would be interesting to be more specific in what your results imply for interpreting the proxy records and derived larger scale reconstructions. And what it would mean, if your results are either too pessimistic or even too optimistic.

Suggestion: Page 8, last paragraph: Are there potentially other reasons that may result in higher correlations, e.g., how the models are built. Did your department's earlier work hint to any further explanations, or did the PAGES project CVAS come up with some additional explanations?

Page 9, line 7: Do I miss it, or do you omit to specify "N".

Suggestion: Page 9, line 18: Maybe make the points of this paragraph already stronger when you present the results.

Suggestion: Page 10, line 1: Can you discuss, how assuming a more appropriate seasonal and depth choice would influence your results?

Page 10, line 1: Isn't the work of Jonkers and Kucera and further of their colleagues relevant here?

Page 10, line 1: I may be wrong, but I think, the work of Jessica Tierney and colleagues on TEX86 calibrations is relevant here.

Conclusion:

Suggestion: Page 11, line 18: I think you could be more explicit about the relevance of your work.

**Technical:**

General: I don't mind seeing "years" written out instead of abbreviated. At least in the abstract I think it would be better to write "400 years" instead of "400y".

Page 1, line 18: Maybe skip "rather"

Page 1, line 20: If I understand the sentence correctly, the second "SNRs" plus its article is superfluous.

Page 1, line 24: I don't think the first sentence of the paragraph is necessary.

Page 4, line 1: If I understand the sentence correctly, it is incomplete.

Page 4, line 7ff: Does this sentence and the next refer to both models or do you mean that you use for TraCE all three mentioned variables? Please clarify.

Page 7, line 19 "differ ..": Please clarify: do you mean they differ between the two simulations?

Page 7, line 20 "consistent with": Please clarify: Do you mean they are consistent with the model or they are consistent in the analyses using this model.
* * *
**Note on Potential Conflicts of Interest:**

As already communicated to the editor prior to agreeing to review:

The journal's guidelines note the following as potential conflicts of interest.

- Reviewers are currently collaborating with an author or have done so recently.
- They have published with an author during the past 3 years.

- They currently hold or have recently held grants with an author.

I and the co-author Laepple are currently part of the same working group and even work package within the German project PalMod (www.palmod.de).

I and the co-author Rehfeld are currently part of the proposal for the follow up phase of PalMod within the same working group and even work package.

I and the co-author Rehfeld co-authored two (non peer-reviewed) meeting reports this year.

Yours sincerely,

Oliver Bothe

---

## Referee Comment (RC2) · Anonymous Referee #2 · 20 Dec 2018

Reschke and colleagues analyze transient climate model simulations in tandem with proxy-data syntheses with the purpose of advancing knowledge on the detectability of climate signals over the Holocene. Their analyses are sound, and the paper is nicely structured and well-written. In general, I am highly supportive of such work, and I certainly consider this manuscript to be worthy of publication in Climate of the Past. However, I have some substantial concerns before I can recommend publication, alongside a few minor comments/questions that might improve the readability and scope of their text.

Substantial Concerns:

1. The assumption of model simulation as reality:

o Although the authors are upfront regarding their underlying assumptions and essentially state that they are taking the model output at face value, I strongly recommend exploring more ways in which the model simulations might be oversimplifying their results. For example, in their Discussion section, the authors discuss the role of the "spatial correlation structure of model simulations" and how these correlations might be overestimated. The authors should add a discussion here about how biases in the simulation of climate variability itself in these transient models can lead to biases in correlation distances.

o In other words, if a (hypothesized) transient simulation from 6 ka to present showed the same, coherent changes across the entire Northern Hemisphere, the calculated SNE, as the authors propose, would be exceedingly low - however, we know that such a transient simulation is an unlikely representation of reality. Thus, I feel that the manuscript would greatly benefit if the authors included text on how typical (and atypical) shortcomings of MPI6k and T21k are influencing their results.

2. Clarity on the separation of "multi-proxy syntheses" versus individual paleoclimate datasets and suggestions for improvement:

o In their abstract, the authors state that "The estimated low signal content of Holocene temperature records should caution against over-interpretation of these kinds of datasets until further studies are able to facilitate a better characterisation of the signal content in paleoclimate records." Here (and later on in their manuscript) the authors need to be very clear about what "these kind of datasets" mean. If they are implying that a broad-brush collation of datasets such as R18 or M13 is over-interpreted, I might agree with them that their analysis tends to demonstrate this aspect. However, this is untrue for a myriad of individual paleoclimate datasets (many of which are subsamples of aforementioned synthesis data sets) that are carefully vetted with high sensitivity to temperature and/or other variables such as precipitation, vegetation, salinity, productivity, etc. and more so, to seasonality - both aspects put together which are not addressed in this paper at all. I strongly recommend rewriting the above statement

in the abstract as well as the final statement in the introduction ("more reliable interpretations of proxy records"; amongst other places) as it unnecessarily detracts from what the authors are proposing. Such statements are also arguably misleading (e.g., modern monitoring and culturing will lead to far better interpretations of proxy datasets compared to estimates of SNR with a climate model) especially considering the point above that their analyses hinge on taking model output at face value.

o The authors' work is a significant advance concerning model-data comparison. In its current version, suggestions on how model simulations or proxy development or the comparison of the two might be improved for better comparative metrics are lacking. I feel that some discussion on how their analyses might be developed further could be helpful.

Minor questions and comments:

- Perhaps I missed it, but why are there no counterpart plots to the T-cent in Fig. 1d-e shown in the main text for T-mill?

- Why does the correlation in T-mill with T21k (Fig. 3e) as well as with Uk'37 and Mg/Ca (Fig. 4b) show an uptick after ∼15000 KM distance?

- What are spatially important regions for proxy record development? Considering that the authors' work is specifically geared towards correlation distances, do their analyses pinpoint which regions are particularly data-deficient (e.g., Indian Ocean, South Atlantic, etc.) and would assist in their comparative metric?

- Is there any particular reason that the authors have not performed a similar analysis with the combined multiproxy datasets of R18, LH14, and M13?

- Again, I would suggest adding up front in the discussion that their analysis explicitly discounts the seasonality of proxies.

- Section 5.1: Is there a reference for anthropogenic forcing strongly increasing correlation decay length? Why necessarily, should this be the case? I feel there ought to be

a statement explaining this here.

- Although the Reschke et al. in review citation is provided, is there any reason for the 1/400y cut-off for the centennial time scale as opposed to something else?

---

## Author Comment (AC1) · 12 Feb 2019

Dear Oliver Bothe,

Thank you very much for taking the time to review our discussion paper and for your constructive and detailed comments. Below we respond to your comments. Referee comments are set in blue italic font and author comments in black normal font.

**Response to Major Comments:**

*Your results and how much we can infer from them strongly hinges on the assumptions you make. You state those clearly and discuss them already. You do this in your discussion section. From my point of view it is necessary that you extend on these discussions already when you present your assumptions.*
*I understand that others may disagree with discussions taking place in the methodssection. Indeed it may be that in a follow up review I say, I was wrong, because the manuscript reads better in this version. However considering this version of the manuscript, the lack of a discussion of your assumptions' appropriateness in the method section clouds the reading.*

*1. Section 3.1: Let me extend on my short summary. First, I think your assumptions are reasonable and well stated and do not invalidate your approach. However, they also can invite strong criticisms. You counter these mostly later in the manuscript, but I think you have to show early that you are thinking about this and why the assumptions are appropriate.*
*A number of questions you should probably deal with early on are, for example: Aren't models thought to be more homogenous than observations? Isn't it unlikely that the proxies really recorded the same 4D-signal? Less important is possibly, whether you can really capture the uncertainty about the signal in the simple estimates you take.*

We agree that it may helpful to add more information on the assumptions earlier in the method section. Our study does not assume the same 4D-signal, but only assumes that the correlation structure in the models is realistic. We argue that simulating a reasonable correlation structure might be easier than simulating the right phase and amplitudes of the climate variability.
As we agree that our results hinges on the correlation structure, we suggest to address this point in the revised version early on by 1.) adding a new section 3.2, discussing the spatial correlation structure in the models vs. the spatial correlation structure in reanalysis data, and 2.) extending the discussion section to include a list of potential model shortcomings that may lead to an overestimation of the spatial coherency in the models (see also our response to your specific comments).
We suggest to leave the remaining part of the discussion (assumption of additive noise) in the discussion section.

Concerning 1.)
To check the realism of the correlation structure in the model simulations, we now further analysed the correlation structure of the surface temperature field in the 20C3M reanalysis product (Compo et al., 2006) (Fig.R1). Interestingly, analysing the full time-period of 1871 to 2011 results in a much higher decorrelation length than estimated for the Holocene, likely caused by the coherent anthropogenic forcing. Removing the last decades to minimise the

human influence, e.g., analysing 1871-1950 results in a correlation structure resembling the spatial correlation of MPI6k.

As we expect that the climate does not get more localised on longer time scales, but if anything, more spatially coherent (e.g., Jones et al., 1997; Kim and North 1991) this suggests that the decorrelation lengths used in this study might not be unrealistically large.

Thus, instead of relying on climate model simulations one could even obtain similar results based on the reanalysis correlation structure and assuming that the correlation structure is similar on longer time scales than on the time scales sampled by the instrumental data. To make this point, we suggest adding the reanalysis correlation structure estimated from the proxy positions in the manuscript Figure 3 (Fig.R2).

One could still argue that fine-scale structures (e.g., at the coast or at shelves) not resolved by the models (as well as by the reanalysis) might lead to localised variations as we already discuss in Section 5.1, but we do not see a clear evidence for this on inter-annual and longer time scales from analysing high-resolution model simulations (e.g., the AWI-FESOM simulation in an eddy-permitting resolution). However, as this latter work is still preliminary we would not include it and just discuss this possibility.

Concerning 2.)

There are several shortcomings in present climate model simulations such as the two simulations used here that may lead to an overestimation of the coherency in the models. Possibilities include that models underestimate internal climate variability that is generally more localised than externally forced climate variability (e.g., Laepple and Huybers, PNAS 2014). One possibility (Laepple and Huybers, GRL 2014) is that the model effective horizontal diffusivity may be too large which would reduce internal variability and lead to larger correlation structures. Further, the low, non-eddy permitting resolution of the model simulations used here might suppress small scale features and the role of persistent coastal currents.

[Figure]

Fig.R1: Decorrelation length of reanalysis data and the 6ky simulation of MPI6k. The decorrelation length is similar for the Holocene and reanalysis data from 1871 to 1950 indicating that the Holocene spatial correlations are realistic.

[Figure]

Fig.R2: Spatial correlation of reanalysis data for the time window from 1871 to 1950 (red lines). As the correlation over distance plots for the reanalysis data are very similar to the ones of the MPI6k this indicates that the spatial correlations of the model data are realistic.

*2. Reference to Reschke et al. (particularly page 6, line 2): Please provide some information in the methods section or at least in an appendix on the method (not least since Reschke et al. is not openly accessible).*

We will add more information in the method section on the reference Reschke et al. to render the manuscript more independent. See our response to the detailed comments below. We further added an unformatted version of the manuscript in the publicly AWI publication database and Researchgate website to allow an easier access to this reference.

**Response to Minor Comments:**

**General:**

*Maybe mention your focus on the last 6kyr already in the title or at least in the abstract.*
We will add this in the abstract.

**Abstract:**

*Suggestion: Maybe rethink the abstract to clarify how you proceed (e.g., page 1, line 17 and following). That is, state which simulations you use before mentioning how they influence the results, or don't mention the specific models at all but just highlight the contrasting results.*
We will mention the specific models in the abstract before explaining the results.

*Suggestion: Mention the use of the correlation structure before discussing its influence.*
We will introduce the correlation structure in the first half of the abstract.

**Introduction:**

Exactly, we consider everything except the specific signal of interest as noise (thus non-climatic signals or climate signals not related to the specific signal of interest). We will rewrite lines 4-10 to make this clearer.

**Data:**

We will follow your suggestion and skip the naming of the 'other' proxies.

We will add the explanation of our annual mean temperature choice. The reason of our choice is to be consistent to the proxy dataset that we assume to record annual mean temperatures following the standard interpretation of these datasets. The annual mean interpretation is often chosen because of the lack of accurate information about the proxy and location-specific seasonality.

Not all forcing factors are continuously transient for the entire T21k simulation. With the disappearance of the Eurasian (~8ky BP) and the Laurentide Ice Sheet (~6ky BP) the transient continental ice sheet forcing ended at around 6ky. As the retreat of the ice sheets ended, there is also no meltwater forcing in the northern hemisphere since 6ky BP. The meltwater fluxes for the southern hemisphere ended at 5ky BP (He, 2011). Thus, only the orbital forcing and the greenhouse gas concentrations are remaining. We will include this in the revised manuscript.

**Method:**

This reference explains the method used for filtering but as we describe the method in the next paragraph, we will remove the reference here.

Our aim is to derive a time series from the annual model time series that resembles the proxy time series in having the same number and ages of the proxy observations. For this, we apply block averaging. To get the observation for the observation time $t_i$ we average all observations between half the difference to the previous observation time ($t_i - \Delta t_i/2$) and half the difference to the next observation time ($t_i + \Delta t_{i+1}/2$). We chose the approach of averaging the annual time series instead of interpolating, as for marine records samples often

include adjacent depths or the sample distance is smaller than the typical mixing depth in the sediment (Berger and Heath, 1968).

We will rephrase the paragraph and include the details of the block averaging in the revised version.

*Suggestion: Page 5, line 16, "For each: : :": I am not sure whether this description gives the reader enough information to redo your analyses. But I am neither sure that it does not. Maybe rethink this.*

We will rewrite this sentence to: 'For each proxy compilation (M13, LH14, R18), we estimated the time scale dependent $(T_{cent}, T_{mill})$ correlations between all possible proxy record pairs. We further estimated the time scale dependent correlations between all model time series pairs.' We further plan to make the R scripts available in an online repository.

*Page 5, line 20: Please give some more details here on what your reference work describes in this context.*

We will extend this description to make it more independent.

'For this step, the irregularly sampled time series were linearly interpolated onto a regular grid ($\Delta t = 10y$) and subjected to a Gaussian filter with cut-off frequency 1/400y ($T_{cent}$) or 1/1000y ($T_{mill}$). This approach has been shown to deliver good results in tests using surrogate data with the sampling properties of Holocene marine sediment cores for the estimation of time scale dependent correlations (Reschke et al., 2019).'

**Results:**

*Suggestion: Page 6, line 25: Would you be willing to discuss how realistic you regard the simulated correlation coefficients, and, possibly, how this may affect the relevance of your SNRs, e.g., if we assume the simulated correlation coefficients are not realistic.*

In the discussion section 5.1 'Spatial correlation structure of model simulations' we already discuss the possibility of overestimating model correlations and their role for the SNRs and we add an additional discussion on potential shortcomings of the climate model simulations. We would like to keep this structure and not to discuss this in the result section. However, as described in our first answer, we suggest to add a new section 3.2 in the method part, discussion the spatial correlation structure in the models vs. the spatial correlation structure in reanalysis data. This demonstrates that model correlations are not unrealistic and that the main conclusions of the manuscript could be obtained without the use of climate models, just relying on the (mainly inter-annual) instrumental correlation structure and assuming that on longer time scales the correlation should not decrease.

*Page 7, line 20: Maybe you should be more explicit in writing about the results for T21k (see also short technical comment below).*

We will rewrite the results of T21k-based estimates to be more explicit and precise.

**Discussions:**

*Suggestion: I think it would be interesting to be more specific in what your results imply for interpreting the proxy records and derived larger scale reconstructions. And what it would mean, if your results are either too pessimistic or even too optimistic.*

We agree that such a discussion would be useful and will add a new discussion section on 'implications and future steps forward'. First, care should be taken in maximising the SNR when creating Holocene climate records. This includes an optimal measurement design (e.g., the choice of the sample size) for example supported by proxy forward modelling. Second, Holocene studies relying on a small number of records might be associated with a large uncertainty (except if these records have a higher quality than the average). Third, Holocene stacks relying on a large number of records, such as the stack in Marcott et al. (2013), will be robust if the errors are independent across sites but it will be very difficult to extract spatio-temporal patterns from these datasets.

If our results are too pessimistic (e.g., the true climate is more regional than simulated by the used model simulations), this would imply that individual proxy records in the Holocene can be safely interpreted as regionally representative climate signal as it is currently done in the literature. On the other hand, if our SNR estimates are too optimistic, the value of singular proxy reconstructions without additional expert knowledge would be very limited and stacks such as used by the tree ring community might be needed.

*Suggestion: Page 8, last paragraph: Are there potentially other reasons that may result in higher correlations, e.g., how the models are built. Did your department's earlier work hint to any further explanations, or did the PAGES project CVAS come up with some additional explanations?*

We agree that it would be useful for the reader to include a discussion on possible model shortcomings that could lead to an overestimation of spatial correlations (= underestimation of spatial degrees of freedom).

Possibilities include that models underestimate internal climate variability that is generally more localised than externally forced climate variability. One suggestion (Laepple and Huybers, GRL 2014) was that the model effective horizontal diffusivity may be too large which would reduce internal variability and lead to larger correlation structures. Further, the low, non-eddy resolving resolution of the models might suppress small scale features and the role of persistent coastal currents. We will add a discussion of these points.

*Page 9, line 7: Do I miss it, or do you omit to specify "N".*

We actually missed to specify N which is the number of sites ranging from 3 to 50. We will add this.

*Suggestion: Page 9, line 18: Maybe make the points of this paragraph already stronger when you present the results.*

We mention this already in page 7, line 23. 'An analysis of the proxy-specific SNRs yielded higher uncertainties due to the relatively small number of record pairs (see Fig. S6-S15 for the complete set of results)' but will make this point clearer and link it to the sensitivity study discussion.

*Suggestion: Page 10, line 1: Can you discuss, how assuming a more appropriate seasonal and depth choice would influence your results?*

Currently, we interpret all records from proxy types as annual mean surface temperature. As different proxies are recording different parts of the climate component, we expect that the correlation among time series from different proxies is lower than for time series of the same proxy which recorded the temperature from a more similar climate component. This is already discussed in lines 3-8.

However, a different season or depth might also have a different correlation structure in the model which will influence our results. Calculating the correlation structure of summer and winter in both models suggests that this can increase or decrease the correlation and seems to be model dependent. Thus, the net-effect on the SNRs is not clear.

Finally, even for one proxy type and proxy carrier (e.g., foraminifera), the recorded season and depth is location-specific and this will reduce the correlation compared to the correlation of the climate sampled at any globally fixed season or depth. However, this reduction in the SNR (that is defined at the moment for annual mean temperatures, but could be changed to any globally fixed season or depth) is real.

We will add a discussion of the latter two points.

*Page 10, line 1: Isn't the work of Jonkers and Kucera and further of their colleagues relevant here?*

This is right. We will add appropriate reference (e.g., Jonkers and Kucera, 2017).

*Page 10, line 1: I may be wrong, but I think, the work of Jessica Tierney and colleagues on TEX86 calibrations is relevant here.*

Our point is that we have seasonal and depth-specific differences in the recorded climate component. In case of TEX86 it was suggested that this proxy records sub-surface temperatures. We will additionally add Tierney and Tingley (2015) proposing a calibration for the upper 200m.

**Conclusion:**

*Suggestion: Page 11, line 18: I think you could be more explicit about the relevance of your work.*

We will add a new section in the discussion on 'Implications and future steps forward' as described above and in the response to reviewer 2. We will further extend/modify this conclusion statement to:

Nevertheless, our SNR estimates are still relevant for synthesis and model comparison efforts (e.g., Marcott et al., 2013), that usually interpret all proxy records together. While in the ideal case, most errors will be averaged out in global stacks based on a large number of records, the interpretation of spatio-temporal patterns will be very uncertain.

**Response to Technical Comments:**

*General: I don't mind seeing "years" written out instead of abbreviated. At least in the abstract I think it would be better to write "400 years" instead of "400y".*

We agree and will change this in the abstract.

*Page 1, line 18: Maybe skip "rather"*

We agree.

*Page 1, line 20: If I understand the sentence correctly, the second "SNRs" plus its article is superfluous.*

We agree.

*Page 1, line 24: I don't think the first sentence of the paragraph is necessary.*
At the moment, one of the shortcoming of our study is that we are not able to make robust statements about specific proxy types as the amount of proxy-specific records is too low as shown in our sensitivity study. In this sense, we would prefer to keep the sentence in the abstract.

*Page 4, line 1: If I understand the sentence correctly, it is incomplete.*
Thanks for spotting this. It is missing an 'and' which we will correct.

*Page 4, line 7ff: Does this sentence and the next refer to both models or do you mean that you use for TraCE all three mentioned variables? Please clarify.*
This refers to both models. We will clarify this.

*Page 7, line 19 "differ .."": Please clarify: do you mean they differ between the two simulations?*
Yes. We will rewrite it to: 'For all three proxy compilations (M13, LH14, R18) the SNRs obtained for mixed proxy types depend on the choice of the model simulation.'

*Page 7, line 20 "consistent with": Please clarify: Do you mean they are consistent with the model or they are consistent in the analyses using this model.*
The latter, we will add that the SNRs estimated for the three datasets M13, LH14 and R18 are more similar and therefore more consistent if the analysis uses the T21k simulation. Based on MPI6k the SNR estimates are more different so that the results are less consistent than the estimates based on T21k.

Once again, thank you for your comments,
Maria Reschke

**References:**

Berger, W. H., and Heath, G. R.: Vertical mixing in pelagic sediments, J. mar. Res., 26, 134-143, 1968.

He, F.: Simulating transient climate evolution of the last deglaciation with CCSM3, PhD thesis, University of Wisconsin-Madison, 2011.

Jones, P. D., Osborn, T. J., and Briffa, K. R.: Estimating Sampling Errors in Large-Scale Temperature Averages, J. Climate, 10, 2548-2568, 1997.

Jonkers, L., and Kucera, M.: Quantifying the effect of seasonal and vertical habitat tracking on planktonic foraminifera proxies, Clim. Past, 13, 573-586, doi:10.5194/cp-13-573-2017, 2017.

Kim, K.-Y., and North, G. R.: Surface Temperature Fluctuations in a Stochastic Climate Model, J. Geophys. Res., 96(D10), 18573-18580, doi:10.1029/91JD01959, 1991.

Laepple, T., and Huybers, P.: Global and Regional Variability in Marine Surface Temperatures, Geophys. Res. Lett., 41(7), 2528-2534, doi:10.1002/2014GL059345, 2014.

Laepple, T., and Huybers, P.: Ocean Surface Temperature Variability: Large Model-Data Differences at Decadal and Longer Periods, P. Natl. Acad. Sci. USA, 111(47), 16682-16687, doi:10.1073/pnas.1412077111, 2014.

Marcott, S. A., Shakun, J. D., Clark, P. U., and Mix, A. C.: A Reconstruction of Regional and Global Temperature for the Past 11,300 Years, Science, 339(6124), 1198-1201, doi:10.1126/science.1228026, 2013.

Reschke, M., Kunz, T., and Laepple, T.: Comparing methods for analysing time scale dependent correlations in irregularly sampled time series data, Comp. Geosci., 123, 65-72, doi:10.1016/j.cageo.2018.11.009, 2019.

Tierney, J. E., and Tingley, M.: A TEX86 surface sediment database and extended Bayesian calibration, Scientific data, 2, 150029, doi:10.1038/sdata.2015.29, 2015.

---

## Author Comment (AC2) · 12 Feb 2019

Dear Reviewer,

Thank you very much for reviewing our discussion paper and your constructive comments. Below we respond to your comments set in blue italic font. The author comments are set in black normal font.

**Substantial Concerns:**

1. **The assumption of model simulation as reality:**

*o Although the authors are upfront regarding their underlying assumptions and essentially state that they are taking the model output at face value, I strongly recommend exploring more ways in which the model simulations might be oversimplifying their results. For example, in their Discussion section, the authors discuss the role of the "spatial correlation structure of model simulations" and how these correlations might be overestimated. The authors should add a discussion here about how biases in the simulation of climate variability itself in these transient models can lead to biases in correlation distances.*
*o In other words, if a (hypothesized) transient simulation from 6 ka to present showed the same, coherent changes across the entire Northern Hemisphere, the calculated SNE, as the authors propose, would be exceedingly low - however, we know that such a transient simulation is an unlikely representation of reality. Thus, I feel that the manuscript would greatly benefit if the authors included text on how typical (and atypical) shortcomings of MPI6k and T21k are influencing their results.*

We agree with the reviewer that including a better discussion of the correlation structure, possible shortcomings in the model simulations and their effect of our results would be useful. This was also asked by Reviewer 1.

We suggest to address this point in the revised version by 1.) adding a new section 3.2, discussing the spatial correlation structure in the models vs. the spatial correlation structure in reanalysis data, and 2.) extending the discussion section to include a list of potential model shortcomings that may lead to an overestimation of the spatial coherency in the models.

Concerning 1.)

To check the realism of the correlation structure in the model simulations, we further analysed the correlation structure of the surface temperature field in the 20C3M reanalysis product (Compo et al., 2006) (Fig.R1). Interestingly, analysing the full time-period of 1871-2011 results in a much higher decorrelation length than estimated for the Holocene, likely caused by the coherent anthropogenic forcing. Removing the last decades to minimise the human influence, e.g., analysing 1871-1950 results in a correlation structure resembling the spatial correlation of MPI6k.

As we expect that the climate does not get more localised on longer time scales, but if anything, more spatially coherent (e.g., Jones et al., 1997; Kim and North, 1991) this suggests that the decorrelation lengths used in this study might not be unrealistically large.

Thus, instead of relying on climate model simulations one could even obtain similar results based on the reanalysis correlation structure and assuming that the correlation structure is similar on longer time scales than on the time scales sampled by the instrumental data. To make this point, we suggest adding the reanalysis correlation structure estimated from the proxy positions in the manuscript Figure 3 (Fig.R2).

One could still argue that fine-scale structures (e.g., at the coast or at shelves) not resolved by the models (as well as by the reanalysis) might lead to localised variations as we already discuss in Section 5.1, but we do not see a clear evidence for this on inter-annual and longer time scales from analysing high-resolution model simulations (e.g., the AWI-FESOM simulation in an eddy-permitting resolution). However, as this latter work is still preliminary we would not include it and just discuss this possibility.

Concerning 2.)
There are several shortcomings in present climate model simulations such as the two simulations used here that may lead to an overestimation of the coherency in the models. Possibilities include that models underestimate internal climate variability that is generally more localised than externally forced climate variability (e.g., Laepple and Huybers, PNAS 2014). One possibility (Laepple and Huybers, GRL 2014) is that the model effective horizontal diffusivity may be too large which would reduce internal variability and lead to larger correlation structures. Further, the low, non-eddy permitting resolution of the model simulations used here might suppress small scale features and the role of persistent coastal currents.

[Figure]

Fig.R1: Decorrelation length of reanalysis data and the 6ky simulation of MPI6k. The decorrelation length is similar for the Holocene and reanalysis data from 1871 to 1950 indicating that the Holocene spatial correlations are realistic.

[Figure]

Fig.R2: Spatial correlation of reanalysis data for the time window from 1871 to 1950 (red lines). As the correlation over distance plots for the reanalysis data are very similar to the ones of the MPI6k this indicates that the spatial correlations of the model data are realistic.

**2. Clarity on the separation of "multi-proxy syntheses" versus individual paleoclimate datasets and suggestions for improvement:**

*o In their abstract, the authors state that "The estimated low signal content of Holocene temperature records should caution against over-interpretation of these kinds of datasets until further studies are able to facilitate a better characterisation of the signal content in paleoclimate records." Here (and later on in their manuscript) the authors need to be very clear about what "these kind of datasets" mean. If they are implying that a broad-brush collation of datasets such as R18 or M13 is over-interpreted, I might agree with them that their analysis tends to demonstrate this aspect. However, this is untrue for a myriad of individual paleoclimate datasets (many of which are subsamples of aforementioned synthesis data sets) that are carefully vetted with high sensitivity to temperature and/or other variables such as precipitation, vegetation, salinity, productivity, etc. and more so, to seasonality - both aspects put together which are not addressed in this paper at all. I strongly recommend rewriting the above statement in the abstract as well as the final statement in the introduction ("more reliable interpretations of proxy records"; amongst other places) as it unnecessarily detracts from what the authors are proposing.*

We agree with the reviewer and will be clearer in the abstract and conclusions to separate between multi-proxy datasets and individual paleoclimate datasets. Specifically, we will precise that 'these kind of data' are large multi-proxy and multi-site data compilations.

*Such statements are also arguably misleading (e.g., modern monitoring and culturing will lead to far better interpretations of proxy datasets compared to estimates of SNR with a climate model) especially considering the point above that their analyses hinge on taking model output at face value.*

We would argue that both methods, case studies such as modern monitoring, culturing, sediment traps, etc. as well as global statistical approaches such as used in this study, will lead to complementary information about the signal contained in the proxy records.

Case studies will be much more precise on the aspects they are analysing, but might omit other effects which are also present in the down-core record. Global statistical approaches include all effects influencing the down-core record, but suffer from the need to make (strong) assumptions.

Ideally, both methods converge to the same results giving credibility that the proxy system and its limitations are completely understood.

*o The authors' work is a significant advance concerning model-data comparison. In its current version, suggestions on how model simulations or proxy development or the comparison of the two might be improved for better comparative metrics are lacking. I feel that some discussion on how their analyses might be developed further could be helpful.*

We agree that this discussion would be useful and will add a new discussion section on implications and future steps forward.

Recent progress in computing power has enabled climate models to perform high resolution, often eddy permitting model simulations (e.g., HighResMIP project) and long simulations (>1000 year) are getting in reach. This is an important step to resolve the spatial scales and regions (mainly shelf areas and coasts) sampled by the proxies. Confronting these results with (replicated) sediment records, ideally accounting for seasonal/depth habitat using heuristic (Jonkers and Kucera, 2017) or complex ecological models (PLAFOM) would allow to better constrain the centennial spatial structures and climate variability as well as to refine the estimates of the proxy signal content shown in this study.

While our assumption of ignoring variations in the seasonal and depth habitat of the proxy recorders and the potential shortcoming in the current model correlation structure might have led to pessimistic SNR estimates, our results still underline the challenge of resolving the small Holocene temperature variations with current marine proxy records. Further improving our understanding of the proxy systems using modern monitoring, culturing and sediment traps and implementing this knowledge into ecological models (Jonkers and Kucera, 2017; PLAFOM) and proxy system models (Dolman and Laepple, 2018) is needed. Forward modelling the proxy records allows to better estimate the signal content and to optimise the sampling (e.g., replication of cores) and measurement process (e.g., sample size, number of foraminiferal tests). Although labour intensive, replicate records would allow to separate local climate variability from non-climate variability and thus provide an important step forward in understanding the proxy and climate variability.

**Minor questions and comments:**

*- Perhaps I missed it, but why are there no counterpart plots to the T-cent in Fig. 1d-e shown in the main text for T-mill?*

We agree that the naming of the figures was misleading. The aim of these figures is to provide a visual impression of the decorrelation lengths based on MPI6k and T21k. To only show one set of maps, we combined here both time scales (time scales larger than 400y, no detrending). The effect of the two time scales can be seen in Figure 3. We will change the nomenclature and describe this more clearly.

- Why does the correlation in T-mill with T21k (Fig. 3e) as well as with Uk'37 and Mg/Ca
(Fig. 4b) show an uptick after 15000 KM distance?

This uptick is likely the result of the orbital forcing that is partly symmetric (effect of obliquity) and antisymmetric (precession) between the hemispheres. For the LH14 dataset (manuscript Fig.3e), the positive correlations at distances >15000km are between the tropics and the northern or southern hemisphere temperate zone as well as between sites of the northern and southern hemisphere temperate zone. There is only one time series pair with negative correlation.

- What are spatially important regions for proxy record development? Considering that
the authors' work is specifically geared towards correlation distances, do their analyses
pinpoint which regions are particularly data-deficient (e.g., Indian Ocean, South
Atlantic, etc.) and would assist in their comparative metric?

There are some regions with a low number of sites such as the Southern Oceans. However, for this kind of study, more important than reducing the lack of single site data in these regions would be enhancing the number of replicate cores (= cores from nearby deployments that were subject to the same climate signal). This would allow to improve estimations of the signal content of proxy records and to test our understanding of proxy formation processes. This is shortly mentioned in 5.1, but we suggest to add this in the new section on implications and future steps forward.

- Is there any particular reason that the authors have not performed a similar analysis
with the combined multiproxy datasets of R18, LH14, and M13?

The datasets were collected with a different focus (M13: reconstruction global temperature; LH14 and R18: temperature variability analysis) and currently use self-consistent, but different calibration and age-modelling approaches. Thus, we use them to test the sensitivity of the results on the choice of the dataset, but combining all datasets would necessitate recalibrations which is beyond our study.

- Again, I would suggest adding up front in the discussion that their analysis explicitly
discounts the seasonality of proxies.

We will add that we neglected in our study the proxy-specific recording and especially seasonality. We will further discuss the effect of ignoring seasonality in more detail as suggested by Reviewer 1.

- Section 5.1: Is there a reference for anthropogenic forcing strongly increasing correlation
decay length? Why necessarily, should this be the case? I feel there ought to be
a statement explaining this here.

The correlation decay length observed in instrumental data (ignoring the last decades) and unforced models is largely consistent with a diffusive energy balance model (Kim and North, 1991) with increasing correlation lengths related to longer time scales (= more time to diffuse). In contrast, forced variability has a correlation length dominated by the spatial pattern of the forcing. For example, for a global forcing such as increasing greenhouse gases this leads to a globally coherent signal overlaying the internal climate variability.

This has been noted by Jones et al. (1997) and to some extend by Sutton et al. (2015), but to our knowledge there is no separate publication on this. However, it is clearly visible when analysing the decorrelation length of the surface temperature field in the reanalysis data (Compo et al., 2006) (Fig.R1). Focussing on the entire reanalysis time period results in a mean

decorrelation length of ~9150km. Contrary, analysing the time window from 1871-1950 the mean decorrelation length is ~3020km (Fig.R1), a finding consistent to the role of anthropogenic forcing.

We will add a short explanation and reference to Jones et al. (1997) in the revised manuscript.

*- Although the Reschke et al. in review citation is provided, is there any reason for the 1/400y cut-off for the centennial time scale as opposed to something else?*

Given a set of time series and their sampling resolution, the optimal cut-off frequency is the highest frequency that can be still resolved by the sampling without introducing a strong bias in the metric of interest, here the correlation.

Simulating surrogate records with the same sampling properties as the true records, Reschke et al. (2019) found that 1/400y is the optimal cut-off for a reasonably large subset of the data used in this study. Due to the Nyquist theorem, one needs at least 2 observations per period, and for typical non-equidistant paleo-data, four times the mean sampling frequency seems to be a rule of thumb appearing from several studies (Laepple and Huybers, PNAS 2014; Reschke et al., 2019) although this will depend on the sampling properties and thus testing this individually using Monte Carlo experiments is the safest option.

Once again, thank you for your comments,
Maria Reschke

**References:**

Compo, G. P., Whitaker, J. S., and Sardeshmukh, P. D.: Feasibility of a 100-Year Reanalysis Using Only Surface Pressure Data, Bull. Amer. Met. Soc., 87, 175-190, 2006.

Dolman, A. M., and Laepple, T.: Sedproxy: a forward model for sediment-archived climate proxies, Clim. Past, 14, 1851-1868, doi:10.5194/cp-14-1851-2018, 2018.

Jones, P. D., Osborn, T. J., and Briffa, K. R.: Estimating Sampling Errors in Large-Scale Temperature Averages, J. Climate, 10, 2548-2568, 1997.

Jonkers, L., and Kucera, M.: Quantifying the effect of seasonal and vertical habitat tracking on planktonic foraminifera proxies, Clim. Past, 13, 573-586, doi:10.5194/cp-13-573-2017, 2017.

Kim, K.-Y., and North, G. R.: Surface Temperature Fluctuations in a Stochastic Climate Model, J. Geophys. Res., 96(D10), 18573-18580, doi:10.1029/91JD01959, 1991.

Laepple, T., and Huybers, P.: Global and Regional Variability in Marine Surface Temperatures, Geophys. Res. Lett., 41(7), 2528-2534, doi:10.1002/2014GL059345, 2014.

Laepple, T., and Huybers, P.: Ocean Surface Temperature Variability: Large Model-Data Differences at Decadal and Longer Periods, P. Natl. Acad. Sci. USA, 111(47), 16682-16687, doi:10.1073/pnas.1412077111, 2014.

Reschke, M., Kunz, T., and Laepple, T.: Comparing methods for analysing time scale dependent correlations in irregularly sampled time series data, Comp. Geosci., 123, 65-72, doi:10.1016/j.cageo.2018.11.009, 2019.

Sutton, R., Suckling, E., and Hawkins, E.: What Does Global Mean Temperature Tell Us about Local Climate?, Phil. Trans. R. Soc. A, 373(2054), 20140426, doi:10.1098/rsta.2014.0426, 2015.